# Molecular Screening of Haemogregarine Hemoparasites (Apicomplexa: Adeleorina: Haemogregarinidae) in Populations of Native and Introduced Pond Turtles in Eastern Europe

**DOI:** 10.3390/microorganisms11041063

**Published:** 2023-04-19

**Authors:** Marko Maričić, Gorana Danon, J. Filipe Faria, D. James Harris

**Affiliations:** 1Institute of Zoology, Faculty of Biology, University of Belgrade, Studentski trg 16, 11000 Belgrade, Serbia; 2BIOPOLIS Program, CIBIO-InBIO, Centro de Investigação em Biodiversidade e Recursos Genéticos, Campus de Vairão, Universidade do Porto, 4485-661 Vairão, Portugal; 3Departamento de Biologia, Faculdade de Ciências, Universidade do Porto, Rua do Campo Alegre s/n, 4169-007 Porto, Portugal

**Keywords:** *Haemogregarina stepanowi*, *Emys orbicularis*, *Mauremys rivulata*, *Trachemys scripta*, Serbia, North Macedonia

## Abstract

Haemogregarines (Apicomplexa: Adeleorina) are the most common and widespread reptilian blood parasites. *Haemogregarina stepanowi* was the first haemogregarine described from a reptile, the European pond turtle *Emys orbicularis*, and initial assessments indicated it was widespread across different pond turtle host species across much of Europe, the Middle East and North Africa. However, recent molecular assessments have indicated the presence of multiple genetically distinct forms in North Africa and the Iberian Peninsula, and extensive mixed infections which may be associated with a negative impact on the hosts. Here, we screened two native species, *E. orbicularis* and *Mauremys rivulata*, and the introduced *Trachemys scripta* from Serbia and North Macedonia for haemogregarines by amplifying and sequencing part of the 18S rRNA gene of these parasites, and used a standard DNA barcoding approach to identify leeches, the final host, attached to pond turtles. Our results again demonstrate the occurrence of considerable diversity of parasites in the analysed pond turtle species, and that *T. scripta* are likely infected by local haemogregarine parasites, and not those that are found in its native range. Leeches were identified as *Placobdella costata,* part of a lineage from Northern Europe. Mixed infections within pond turtles were again common. Current haemogregarine taxonomy does not reflect the genetic diversity identified, and a full taxonomic reassessment is needed.

## 1. Introduction

The importance of parasites, both individually and in terms of ecological interactions, was historically underestimated by most ecologists [1]. Although the significance of parasites is now better appreciated, various aspects remain underassessed and this is particularly true for taxonomic diversity, with approximately only 0.1% of apicomplexans—a diverse group of obligate intracellular parasitic organisms—having been described [2]. Abundance and distribution data, in terms of both the geographic distribution patterns of parasites and the different hosts in which they occur, need to be prioritized to allow improved conservation decision making and to identify the processes that shape parasite diversity [3].

Haemogregarines (Apicomplexa: Adeleorina) are the most common and widespread reptilian blood parasites [4]. After an extensive taxonomic reorganization, Siddall [5] recognized 19 species in the genus *Haemogregarina*, all in turtle intermediate hosts with leeches as the final hosts. Since then, molecular data has led to the transfer of some species to other genera [6], and indicated that *Haemogregarina* also occur in other vertebrates exposed to leeches, such as fish [7].

*Haemogregarina stepanowi* was the first haemogregarine described from a reptile, in the European pond turtle *Emys orbicularis*. Molecular data indicate *H. stepanowi* is geographically widespread in this species, from Syria and Iran to Algeria, Bulgaria and Sicily, where it infects the closely-related Sicilian pond turtle *Emys trinacris* [8], as well as occurring in three other western Palearctic terrapins—*Mauremys leprosa*, *Mauremys caspica* and *Mauremys rivulata* [9]. However, recent assessments have highlighted the presence of multiple, genetically distinct forms in *M. leprosa* in Morocco [10] and Tunisia [11] and in *E. orbicularis* in Tunisia [11] and the Iberian Peninsula [12]. In *M. caspica*, an additional form was also reported, which showed close morphological similarities with *H. stepanowi*, but which was genetically highly distinct [13]. Information regarding the distribution and host ranges of these lineages is crucial for disease management, because in places where multiple parasite forms occur, mixed infections have been reported [11,12,14] and linked to lower values of haematocrit, which was not identified in hosts infected by a single parasite lineage [15].

In this study, we screened 130 terrapin individuals of three species from Serbia and North Macedonia: the native *E. orbicularis* and *M. rivulata* and the introduced *Trachemys scripta*. Our aims were: to assess parasite diversity in *E. orbicularis* from this region to compare to recent data from the Iberian Peninsula and North Africa; to assess the possibility of mixed infections; and to determine if there was any evidence of host specificity among the parasite lineages detected. Since haemogregarine lineages within *T. scripta* have previously been identified in its native America [16], we can further assess potential parasite spill-over between an invasive exotic species and the native species.

## 2. Materials and Methods

Samples were collected from 7 different localities in Serbia and North Macedonia (Figure 1) during April to August 2021. Turtles in the wild were caught by hand or with baited hoop traps. Blood samples were collected with syringes and stored in 1.5 mL cryotubes containing 1 mL of 96% ethanol. During collection, several terrapins had leeches attached, and these were removed and also stored in 96% ethanol for later genetic approaches. For DNA extraction we followed a standard saline protocol, as in [17]. For genetic analysis, 130 terrapins were screened using PCR, using the primers HepF300 and HepR900 to amplify part (approximately 600 bp) of the 18S rRNA gene [18]. These primers were designed to amplify haemogregarines of the genus *Hepatozoon* and have been used successfully to amplify haemogregarines from pond turtles [10,15], but are also known to amplify other endoparasites [19]. PCR reactions were run in a reaction mixture containing 1.5 μL MgCl_2_, 0.4 μL of each nucleotide, 2 μL of PCR buffer (200 mM Tris-HCl, 500 mM KCl), 1 μL of each primer, 0.4 mg/mL of albumin (BSA; Roche, Switzerland), 2 μL of DNA and 0.1 μL of Taq DNA polymerase and filled to a total volume of 20 μL with pure water. Negative controls were included in each PCR, with the DNA replaced with sterile water. The PCR reaction mix was heated to 98 °C for 10 min, and amplification was performed for 40 cycles at 98 °C for 30 s, 60 °C for 45 s, and 72 °C for 45 s, followed by a final 10 min extension at 72 °C. To confirm the identity of the leeches, eleven individuals from separate pond turtle hosts including both *E. orbicularis* and *M. rivulata* were assessed. DNA was extracted in the same way as for the pond turtles, and a section of the mitochondrial COI gene was amplified using the universal primers LCO1490 and HCO2198 [9]. The success of the PCR was assessed by electrophoresis of 2 μL of sample in a 2% agarose gel, visualized by staining with GelRed (Biotarget, Portugal) under UV light. Positive PCR products of the expected size for both haemogregarines and leeches were purified and sequenced by a commercial sequencing entity (Genewiz, Germany).

A preliminary identification of sequences was made by comparison against published sequences on GenBank using BLAST [20]. Regarding the haemogregarines, several electropherograms demonstrated clean, readable sequences that became unreadable after positions of known length variants between separate phylogenetic lineages of haemogregarines. These were interpreted as mixed infections, as has been done in previous studies (e.g., [12]). The remaining, fully readable sequences of haemogregarines (GenBank Accession Numbers OQ799319 to OQ799344) were aligned with published data from GenBank in Geneious Prime 2021.1.1 (Biomatters Ltd., Auckland, New Zealand) using clustalW. Two sequences from *Hemolivia* (*H. stellata* Petit, Landau, Baccam and Lainson, 1990 and *H. mauritanica* (Sergent and Sergent, 1904)) were included as outgroups. The resulting alignment included 89 sequences, with a total length of 552 base pairs. For the leeches, all the most similar sequences available on GenBank corresponded to *Placobdella costata*. These were aligned against the new sequences (GenBank Accession Numbers OQ799299 to OQ799309), with two sequences from *Placobdella parasitica* included as outgroups. The aligned dataset of CO1 sequences consisted of 62 sequences with a total length of 626 base pairs. As expected for a protein coding gene, no insertions or deletions were identified, and all sequences could be translated without unexpected stop codons. For the phylogenetic analyses, PartitionFinder v2.1.1 [21] was used first to select the most appropriate model of molecular evolution under the AICc, and then a phylogeny was estimated for each dataset through Bayesian Inference (BI) using MrBayes v3.2 [22]. The selected models were the GTR + I + G (haemogregarines) and JC for starting position 1, and SYM + G for starting positions 2 and 3 (leeches), and in both cases the BI was run for 3 × 10^7^ generations, with a sampling frequency of 1000 and all other parameters left as default.

## 3. Results

Of the 130 pond turtles screened for haemogregarines, 26 yielded fully readable sequences, and provided the closest matches with species of *Haemogregarina* using BLAST. These included two samples of *T. scripta*, five of *M. rivulata*, and nineteen samples of *E. orbicularis*. Eight further samples gave evident mixed infections, including five *E. orbicularis* and three *M. rivulata*. A sequence from a single specimen (EMC89) was apparently from an unidentified stramenopile, with a 98.8% match with an uncultured biosoecida from China (FJ4105251). Phylogenetic analysis of the haemogregarines (Figure 2) indicated that multiple distinct species were present among the 26 usable samples that were included. Two lineages were identified in both *T. scripta* and *M. rivulata*, while lineages from *E. orbicularis* were distributed across three major groups (Figure 2).

Sequences were generated from all 11 leeches, all of which had BLAST matches with 99% or more with *Placobdella costata*. Estimates of phylogenetic relationships of the CO1 sequences (Figure 3) identified the same seven major lineages (labelled as C1–C7) previously described [23]. All of the leeches sequenced in this study were part of the clade identified as “C4” in [23], which includes specimens from Latvia and Germany, but also a specimen from the Saratov Oblast in Russia [24].

## 4. Discussion

Phylogenetic analyses indicate that multiple distinct parasite lineages were found within each of the three terrapin host species examined. In *E. orbicularis* from the Iberian Peninsula, parasites were identified as belonging to three major groups (labelled A–C in [12] and in Figure 2), and after including our new data from Eastern Europe the same pattern was observed. Seven of the new positive samples of *E. orbicularis* formed part of group C, and these were identical or almost so to *Haemogregarina stepanowi*, and so can tentatively be identified as this species. However, other named species such as *H. balli* are also part of this group. Five other new samples from *E. orbicularis* hosts fell within group B, along with other samples from *E. orbicularis* and *M. leprosa* from the Iberian Peninsula, Menorca and Morocco. Seven other samples likewise form part of group A, also including samples from *E. orbicularis* and *M. leprosa* from the Iberian Peninsula and Morocco. In the new data from *M. rivulata*, two distinct lineages were identified—three haplotypes in group C, possibly corresponding to *H. stepanowi*, and two in group D, along with sequences from *M. leprosa* in Morocco and *Mauremys caspica* from Iran (Figure 2). Finally, two new haplotypes were identified from *T. scripta* hosts, one forming part of group A, and the other part of group B. In its native range, several different haplotypes of parasites have been previously identified from *T. scripta* hosts [25], but the two haplotypes identified in this study are similar to those from other European pond turtles, and not those from the Americas.

These results illuminate various aspects of the patterns of diversity of parasites of the genus *Haemogregarina* with respect to their terrapin hosts, but also identify facets that require further investigation. One of the negative aspects of invasive species is the possibility that they also spread parasites to native, potentially naïve, hosts. The alien leech *Helobdella octatestisaca* has been identified in Spain attached to pond turtles, possibly associated with introduced aquatic plants [26]. *Trachemys scripta* is often considered the most widely invasive reptile species in the world [27], and host-switching of flatworm endoparasites from introduced *T. scripta* to native *M. leprosa* has already been identified in natural environments in the Mediterranean region [25]. On the other hand, haemogregarines were rare in captive animals in the USA [16], so when *T. scripta* are introduced from captive populations, the risk of introduction of blood parasites may be low. Not only that, but it also seems that the *T. scripta* are being infected by local haemogregarine parasites, and not those that are found in the native range. Since the impact of the different parasites remains unknown, this could be an appropriate system to study the effect of native and non-native parasites on the vertebrate host.

Where large numbers of pond turtles have been screened for haemogregarines, whether in Iberia [12], Morocco [28], Tunisia [11] or as in this study in Eastern Europe, multiple highly distinct lineages have been recognized. This seems at odds with early studies, which identified a single, widespread parasite, *H. stepanowi*, across much of Europe and North Africa (e.g., [9]). It seems likely that not all primers used in molecular studies reliably amplify the different lineages, and indeed recent studies have highlighted this issue [18,29]. This may also explain why mixed infections were only relatively recently identified [11,15], while in fact they seem to be relatively common. It has been suggested that identification of haemogregarines based on morphology of gamonts is difficult, while sequencing of the 18S rRNA gene can provide an important additional characteristic [30]. This is much more problematic if some primers do not amplify diverse lineages, and this aspect therefore clearly deserves further investigation. On the other hand, the primers used in this study amplify across related genera including *Hepatozoon*, *Karyolysus* and *Hemolivia* [31,32]. Therefore, these may be a good option for future molecular screening approaches until other possible primers are fully assessed. The identification of multiple distinct lineages within single host specimens also causes difficulties in interpreting the electropherograms resulting from direct PCR and Sanger sequencing approaches, in particular when there are length variants that cause the different sequences to be out of phase. It is sometimes possible to compare heterozygous positions and the locality of the length variant within the sequences, and directly infer which haplotypes are present [12]. An alternative approach would be to clone and then resequence when mixed infections within an individual are identified.

Information regarding the distribution of different lineages of parasites is essential to better determine the impact of parasites on the pond turtle hosts. Haemogregarine blood parasites are typically considered to be benign, but the impact of mixed infections of blood parasites on the hosts remains largely unknown, since most studies identifying them have been purely descriptive (reviewed in [4]). Infection by *H. stepanowi* can induce erythrocytic changes, giving a flattened aspect to the nucleus, while in erythrocytes infected with trophozoites the nucleus can be displaced [33], but it is not clear what impact this has on the host. An assessment of 30 *E. orbicularis* seized in transit through Serbia indicated that all animals were in poor health, with necrotic ulcerations and skin hemorrhages, and all animals were infected with haemogregarines [34]. However, these authors also note that the *Haemogregarina* infection may not be correlated with the health condition of these animals, and other studies have reported 100% prevalence of infections in pond turtles with no clinical symptoms [35]. Likewise, although these 30 animals infected with *Haemogregarina* had lower values of haematocrit compared to reference values (values from [34] compared to [36]), it is not clear whether there is a causal relationship between the infection and the low haematocrit values obtained. In a study of *M. leprosa* in the Iberian Peninsula, there were no significant differences in body condition between uninfected turtles and those with single or mixed lineage infections, but individuals with mixed infections had the lowest values of haematocrit [15]. Furthermore, immune response varied, with individuals with mixed infections having a higher immune response than those uninfected or infected by a single lineage. These findings were considered to demonstrate the negative effects of mixed infections [15]. This negative impact, and the current findings that mixed infections are much more common that previously perceived, highlights the need for a detailed assessment of this aspect.

Differences in prevalence of *Haemogregarina* infections have been widely reported both between geographic regions and within small areas. In *Emys trinacris* from Sicily, 9% of individuals were infected, with positive individuals identified in two out of eleven sampled populations [8]. On the other extreme in a population of *E. orbicularis* from Romania 100% of individuals from one population were infected [35]. Similarly, 30 out of 30 individuals seized in transit in Serbia and examined in Belgrade zoo were infected [34]. In two populations of each of *E. orbicularis* and *M. caspica* in Iran, 73% and 58% were infected, respectively [37]. In this study, six of seven populations of *E. orbicularis* had infected individuals, with prevalence ranging from 0–75%, although, as in many of the previous studies, the number of sampled individuals in some populations was low. Reasons for these large differences in prevalence remain unclear. A recent assessment of three populations of *M. rivulata* in Turkey reported that 54% of individuals were infected with *H. stepanowi* [38]. The population with the highest water pollution had the highest infection rate (90%), but also the largest animals. Larger adults present a better target for leech attachment because of their larger surface area, and furthermore pond turtles with a longer life span are expected to be more exposed to leeches [39], and therefore are expected to have greater exposure to haemogregarines.

Concerning the final host, the leech *Placobdella costata*, all individuals sequenced in this study were almost identical to previously sequenced specimens from Latvia and Germany [23]. These authors have suggested that *P. costata* may be considered an incipient species complex, with seven independent lineages potentially corresponding to seven distinct species (Figure 3, Labelled C1–C7). A distinct lineage was found in Montenegro (C7), but there was previously no information regarding which lineage occurred in Serbia or North Macedonia. Interestingly, in the study of [23], specimens from Russia, Ukraine, Bulgaria and Turkey all formed a single lineage (C6). However, our inclusion of an additional specimen from the Saratov Oblast, Russia [24] shows that this specimen is found in the same clade as our samples, and those from Latvia and Germany. Our data therefore helps clarify the distribution of these leech lineages, but also demonstrates that the distribution of forms may not be as geographically discrete as previously considered. This will have taxonomic implications if these lineages are formally named, since the type specimen of *P. costata* is from Ukraine, but it is no longer evident if only one lineage is found in this region. The absence of haemogregarines in populations of pond turtles south of the Atlas Mountains in Morocco was linked to a lack of leeches in this area [10]. However, more detailed attempts to model leech distributions along with those of haemogregarines have been hampered by limited distribution data, and biases in sampling points [29]. Given that *P. costata* is the only European leech considered capable of feeding on reptile blood [40], it is evident that a single final host can harbour many different species of haemogregarine, which will further complicate modelling approaches, but may well explain the lack of specificity in the terrapin hosts in this system. Until recently, in Europe, *P. costata* was considered to be always associated with *E. orbicularis*, with only occasional records from *M. leprosa* [41]. However, recent assessments in Morocco identified *P. costata* regularly attached to *M. leprosa* [28], and it has also been reported attached to *M. caspica* in northern Iran [42]. In this study this leech was found attached to *M. rivulata* as well as *E. orbicularis*. The European range of *E. orbicularis* has greatly reduced in the last century, particularly due to anthropogenic habitat changes such as drainage of marches and intensified agriculture, leading to the suggestion that this may force the leech to find other hosts, including possibly amphibians and birds [39]. At the same time, pond turtles such as *M. leprosa* and *Mauremys sinensis* have been introduced [43] that may be potential alternative hosts. Elucidation of *P. costata* distribution and host use is clearly needed to also shed light on the distribution of associated haemogregarines.

Stramenopiles have been amplified in reptile tissue samples before with this same molecular approach, notably blood parasites of the genus *Proteromonas* [44,45]. In this study, a single *E. orbicularis* specimen yielded a sequence for which the closest comparative sequence belongs to a bicosoecida, a group of free-living unicellular flagellates. With the minimal comparative sequence data available, no concrete conclusions can be drawn regarding what this sequence may correspond to, but by reporting the sequence (GenBank Accession number OQ801489), it may be valuable for future assessments concerning the distribution and diversity of these poorly known organisms.

To conclude, we carried out the first comprehensive assessment of blood parasite diversity within *E. orbicularis* from Serbia and North Macedonia, and, to the best of our knowledge, the first molecular screening of introduced populations of *T. scripta*. When combined with previous assessments in North Africa and Iberia, our results confirm a pattern of high diversity and extensive mixed infections in this host. Our evaluation of *M. rivulata* and *T. scripta* from the same areas shows no indication of host specificity for these parasites. The leech final host, *P. costata*, is placed in a phylogeographic framework, and as the likely sole host for all the *Haemogregarina* species present, indicates that many different blood parasite species can employ the same leech host. We found no indication of introduced blood parasites from the invasive *T. scripta*, but rather locally common haemogregarines are now using this alien species as a host. It is evident that current haemogregarine taxonomy does not reflect the genetic diversity identified, and a full taxonomic reassessment is needed. However, the fact that no new lineages of parasites were identified may suggest that much of the diversity of these blood parasites in European and North African pond turtles has now been recognized, but awaits formal naming after inclusion of morphological data.

## Figures and Tables

**Figure 1 microorganisms-11-01063-f001:**
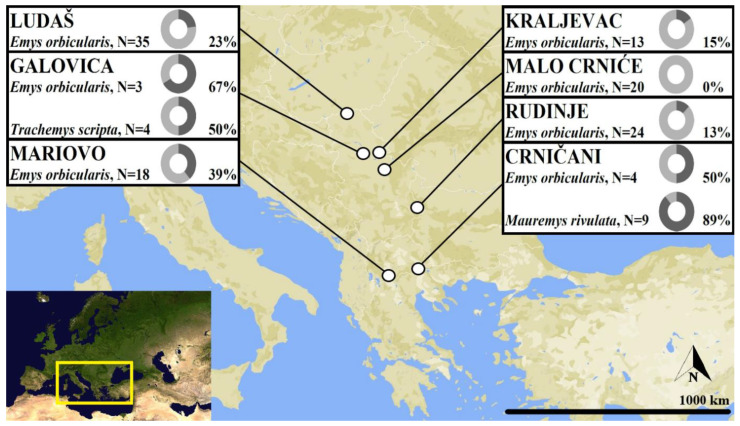
Map showing the sampling localities of pond turtles. Insets show the number of each host species assessed, and the percent infected with haemogregarines, as determined by PCR and DNA sequencing.

**Figure 2 microorganisms-11-01063-f002:**
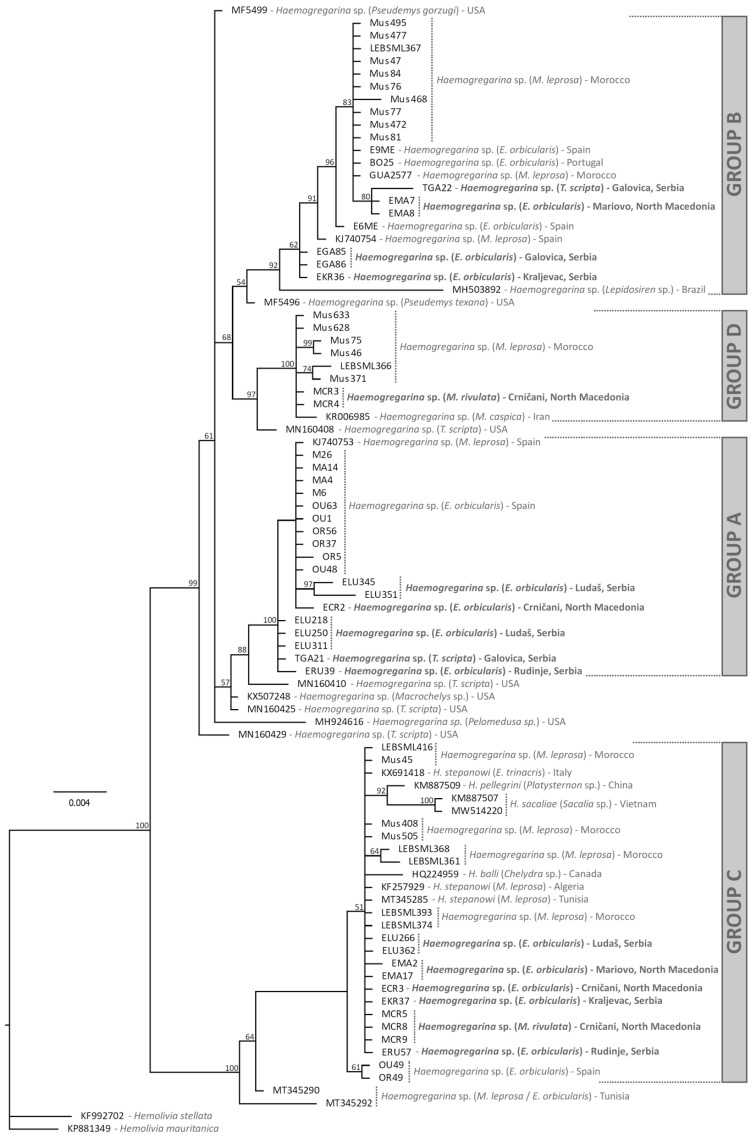
Estimate of phylogenetic relationships of the *Haemogregarina* detected based on Bayesian inference of partial 18S rRNA sequences. Values next to nodes indicate % Bayesian Posterior Probabilities. The tree was rooted using two examples from the genus *Hemolivia*.

**Figure 3 microorganisms-11-01063-f003:**
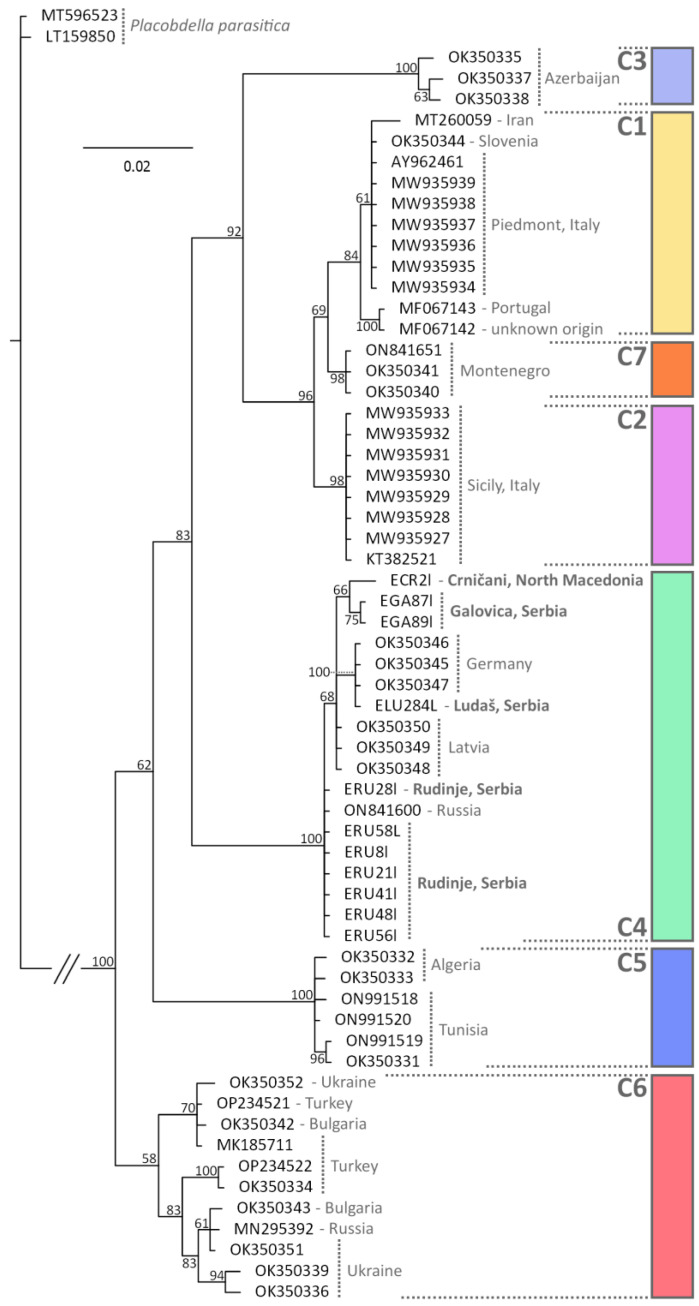
Estimate of phylogenetic relationships between lineages of *Placobdella costata* based on Bayesian inference of partial CO1 mitochondrial DNA sequences. Values next to nodes indicate % Bayesian Posterior Probabilities. Lineages (C1–C7) correspond to those previously identified by [23]. The tree was rooted using two examples of *Placobdella parasitica*.

## Data Availability

All sequences are available on GenBank as indicated in the text.

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
