# Peer review of "Molecular Screening of Haemogregarine Hemoparasites (Apicomplexa: Adeleorina: Haemogregarinidae) in Populations of Native and Introduced Pond Turtles in Eastern Europe"

_microorganisms, 2023, doi:10.3390/microorganisms11041063_

Round 1
Reviewer 1 Report
In the present study, 6 out of 7 populations of E. orbicularis had infected pond turtles, with prevalence ranging from 0-75%, but unfortunately, the number of individuals sampled in some populations was low(n=3-4).
It is necessary to specify in the text, the material and method, more precisely the area of origin of the turtles, because Eastern Europe is too extensive area compared to their real origin.
In the discussion part, the relationship between the results of the study and those in the literature must be shown more clearly.
The conclusions should be more focused on the importance of the research results and their originality, novelty.
It is necessary to expand and update the bibliographic list, considering the wealth of sources, having the following examples:
MIHALCA,et all., Effect of Haemogregarina stepanovi Danilewsky, 1885 (Apicomplexa: Haemogregarinidae) on erythrocyte morphology in the european pond turtle, Emys orbicularis (Linnaeus, 1758) Blanford, 1876 (Testudines: Emydidae). Scientia Parasitologica, 2004, 1-2, 175-179
Al-Quraishy, S.; Abdel-Ghaffar, F.; Dkhil, M.A.; Abdel-Gaber, R. Haemogregarines and Criteria for Identification. Animals 2021, 11, 170. https:// doi.org/10.3390/ani11010170
Jòzsef, Ö.; Darko, M.; Milos, V.; Bojan, G.; Jevrosima, S.; Dejan, K.; Sanja, A.-K. Cytological and molecular identification of Haemogregarina stepanowi in blood samples of the European pond turtle (Emys orbicularis) from quarantine at Belgrade Zoo. Acta Vet.-Beograd 2015, 65, 443–453.
Rakhshandehroo, E.; Sharifiyazdi, H.; Ahmadi, A. Morphological and molecular characterisation of Haemogregarina sp. (Apicomplexa: Adeleina: Haemogregarinidae) from the blood of the Caspian freshwater turtle Mauremys caspica (Gmelin) (Geoemydidae) in Iran. Syst. Parasitol. 2016, 93, 517–524
Author Response
In the present study, 6 out of 7 populations of E. orbicularis had infected pond turtles, with prevalence ranging from 0-75%, but unfortunately, the number of individuals sampled in some populations was low(n=3-4).
It is necessary to specify in the text, the material and method, more precisely the area of origin of the turtles, because Eastern Europe is too extensive area compared to their real origin.
Reply: This has been changed both in the material and method and in various places throughout the manuscript.
In the discussion part, the relationship between the results of the study and those in the literature must be shown more clearly.
Reply: In the first paragraph of the discussion we now stress which are the new sequences derived from this study, and which were previously published
The conclusions should be more focused on the importance of the research results and their originality, novelty.
Reply: We have rewritten the beginning of the concluding paragraph to stress this better.
It is necessary to expand and update the bibliographic list, considering the wealth of sources, having the following examples:
MIHALCA,et all., Effect of Haemogregarina stepanovi Danilewsky, 1885 (Apicomplexa: Haemogregarinidae) on erythrocyte morphology in the european pond turtle, Emys orbicularis (Linnaeus, 1758) Blanford, 1876 (Testudines: Emydidae). Scientia Parasitologica, 2004, 1-2, 175-179
Al-Quraishy, S.; Abdel-Ghaffar, F.; Dkhil, M.A.; Abdel-Gaber, R. Haemogregarines and Criteria for Identification. Animals 2021, 11, 170. https:// doi.org/10.3390/ani11010170
Jòzsef, Ö.; Darko, M.; Milos, V.; Bojan, G.; Jevrosima, S.; Dejan, K.; Sanja, A.-K. Cytological and molecular identification of Haemogregarina stepanowi in blood samples of the European pond turtle (Emys orbicularis) from quarantine at Belgrade Zoo. Acta Vet.-Beograd 2015, 65, 443–453.
Rakhshandehroo, E.; Sharifiyazdi, H.; Ahmadi, A. Morphological and molecular characterisation of Haemogregarina sp. (Apicomplexa: Adeleina: Haemogregarinidae) from the blood of the Caspian freshwater turtle Mauremys caspica (Gmelin) (Geoemydidae) in Iran. Syst. Parasitol. 2016, 93, 517–524
Reply: the third of these references was already cited (ref 30 in previous version, now 34 – the names as written by the reviewer are incorrect, but the reference is the same). We have added the remaining three references to the text, as well as 2 other new references.
Reviewer 2 Report
Summary
This manuscript reports on a survey of hemogregarines in turtles in seven ponds in Serbia and North Macedonia in the summer of 2021. Blood samples and leeches were collected from 130 turtles comprising 2 native and 1 invasive North American species. 18S rDNA PCR amplification from blood samples using primers designed for hemogregarines (but with a reportedly able to amplify a wider range of protozoa, notably Coccidia) followed by Sanger sequencing and database comparison and phylogenetic analysis. A similar approach was applied to the leeches. 26 blood samples provided unique sequence, which basically fell into 4 different known groups of hemogregarines. No American parasites introduced with the turtle were detected, which were infected with European parasites. This is consistent with the notion that these turtles were released pets, and it is known that turtles bred in captivity are free of hemogregarines. In essence, this does not pose a risk on the native turtles. The leeches were nearly identical to sequences detected widely in Europe and North Africa and provided some further insights on the wider geographic ranges of leech species and sub-types. Overall, this survey consolidates previous observations at different sites and times, but also expands the insights on leech distribution. Overall, the study is well executed and well-presented and discussed.
Specific points
1. The eight samples with unreadable sequence were interpreted as mixed infections. It would have been nice to have cloned these and then sequenced to reveal the mixed species. In particular, if this could have been an American parasite in the mix, which is now left somewhat unresolved.
2. Although the specificity or species range of the 18S rDNA primers is debated, and this primer pair seems wide-catching, it is an unknown and would certainly warrant further scrutiny in future experiments. Currently, this seems the accepted primer pair in the field, but it is not without significant question marks on its effectivity.
Author Response
This manuscript reports on a survey of hemogregarines in turtles in seven ponds in Serbia and North Macedonia in the summer of 2021. Blood samples and leeches were collected from 130 turtles comprising 2 native and 1 invasive North American species. 18S rDNA PCR amplification from blood samples using primers designed for hemogregarines (but with a reportedly able to amplify a wider range of protozoa, notably Coccidia) followed by Sanger sequencing and database comparison and phylogenetic analysis. A similar approach was applied to the leeches. 26 blood samples provided unique sequence, which basically fell into 4 different known groups of hemogregarines. No American parasites introduced with the turtle were detected, which were infected with European parasites. This is consistent with the notion that these turtles were released pets, and it is known that turtles bred in captivity are free of hemogregarines. In essence, this does not pose a risk on the native turtles. The leeches were nearly identical to sequences detected widely in Europe and North Africa and provided some further insights on the wider geographic ranges of leech species and sub-types. Overall, this survey consolidates previous observations at different sites and times, but also expands the insights on leech distribution. Overall, the study is well executed and well-presented and discussed.
Reply: Thanks for the positive feedback
- The eight samples with unreadable sequence were interpreted as mixed infections. It would have been nice to have cloned these and then sequenced to reveal the mixed species. In particular, if this could have been an American parasite in the mix, which is now left somewhat unresolved.
Reply: This is correct, although in some previous cases it has been possible to decipher mixed infections by examining exactly which positions are heterozygous, and where sequences go out of phase (eg ref 16). However, as more haplotypes are known this becomes harder to do, and in our case we were not able to infer haplotypes with any certainty. We agree, cloning would be an option and add some text about this (Lines 222-228)
- Although the specificity or species range of the 18S rDNA primers is debated, and this primer pair seems wide-catching, it is an unknown and would certainly warrant further scrutiny in future experiments. Currently, this seems the accepted primer pair in the field, but it is not without significant question marks on its effectivity.
Reply: The primers used in this study amplify related genera such as Hepatozoon, Karyolysus and Hemolivia, as well as occasionally amplifying distantly related taxa such as the bicosoecida identified in this manuscript. It seems therefore that these are more likely to amplify all lineages, rather than some primers used in other studies that were modified or designed for specific groups. We agree that this is an issue that needs further investigation, and have added some text regarding this (Lines 215-222).